# How Similar Are Proteins and Origami?

**DOI:** 10.3390/biom12050622

**Published:** 2022-04-21

**Authors:** Hay Azulay, Aviv Lutaty, Nir Qvit

**Affiliations:** 1Independent Researcher, Koranit 2018100, Israel; 2Independent Researcher, Kiryat Motzkin 2641312, Israel; 3The Azrieli Faculty of Medicine in the Galilee, Bar-Ilan University, Henrietta Szold St. 8, POB 1589, Safed 1311502, Israel

**Keywords:** protein, origami, tessellation, tensegrity, auxetic

## Abstract

Protein folding and structural biology are highly active disciplines that combine basic research in various fields, including biology, chemistry, physics, and computer science, with practical applications in biomedicine and nanotechnology. However, there are still gaps in the understanding of the detailed mechanisms of protein folding, and protein structure-function relations. In an effort to bridge these gaps, this paper studies the equivalence of proteins and origami. Research on proteins and origami provides strong evidence to support the use of origami folding principles and mechanical models to explain aspects of proteins formation and function. Although not identical, the equivalence of origami and proteins emerges in: (i) the folding processes, (ii) the shape and structure of proteins and origami models, and (iii) the intrinsic mechanical properties of the folded structures/models, which allows them to synchronically fold/unfold and effectively distribute forces to the whole structure. As a result, origami can contribute to the understanding of various key protein-related mechanisms and support the design of de novo proteins and nanomaterials.

## 1. Introduction

Proteins have been the focus of many research programs for over 70 years, yet their folding mechanism, structural properties, and function are still not fully understood. Origami, the art of folding a two-dimensional (2D) paper into a three-dimensional (3D) model, dates back to the Edo period (1603–1867) as a traditional Japanese craft for religious ceremonies and games. In recent years, origami folding principles and models have been implemented in a range of disciplines, including mathematics, physics [1,2], material engineering [3], mechanical engineering [4], architecture [5], and biology [6]. For example, engineers design innovative origami-based mechanical devices and structures [7,8], and since folding is interwoven into various natural phenomena, including the folding of leaves, flowers, and wings of birds, biologists seek inspiration in origami for novel models and analysis tools [9].

There are obvious similarities between origami and protein folding, the physical process by which a protein chain acquires its native 3D structure. For example, in both cases, the creation of a structure involves sequential folding through a series of steps [10] (Figure 1). Furthermore, various biomolecular structures can be folded from a single chain of amino acids, just as many origami models can be formed from a single sheet of paper. Hence, it is not surprising that protein folding is known as “Nature’s Origami” [11]. Another method to construct 2D and 3D molecules from DNA is called “DNA origami” [12], and it is similar to the assembly of folded subunits in modular origami. Understanding the folding process is important since misfolding can lead to undesired inactive proteins, parallel to an undesired model in origami. Even worse, misfolding can result in malfunctioning proteins, which can lead to various diseases (e.g., neurodegenerative disorders, diabetes, and cardiovascular diseases). 

Researchers have attempted to harness origami folding principles to understand the folding process of proteins [10]. However, although it may seem trivial to experts in the field of protein or origami research, no study has holistically compared proteins and origami folding processes, structures, and mechanical properties. Such analysis can provide new insight into miscellaneous biomolecular structures and complex origami models. This article covers the evidence for the similarity between proteins and origami from physics, mathematics, chemistry, biology, and structural research. The topics that are discussed include folding constraints, continuity of the folded medium, shape/structure, and mechanical properties of protein structures and origami models. The results demonstrate consistent similarities between the initial primary material features, the creation process, and the properties of the final structures of proteins and origami models, indicating that origami can be an effective reference for studying proteins.

## 2. Folding Processes 

The motivation for understating the mechanism of protein folding is obvious. From a basic research perspective, protein structure is highly important for the function it fulfills (e.g., channels and receptors) and affects its interactions with other proteins and molecules. Therefore, understanding how proteins fold can lead to a better prediction of the structure that would fold from an amino-acid chain, and it may guide the design of de novo proteins, with predetermined structures and functions. Aberrant protein folding is believed to be the primary cause of various pathologies that are categorized into two major groups: loss-of-function (e.g., cystic fibrosis and a wide range of metabolic defects), and toxic gain-of-function, in which metastable protein aggregates are associated with cellular toxicity (e.g., neurodegenerative diseases, cardiovascular disease, and cancer). The complex process of protein misfolding can lead to cytotoxicity, subversion of tissue architecture, progressive organ damage, and eventually death, and it is believed to be the primary cause of Alzheimer’s disease [13], Parkinson’s disease [14], as well as prion diseases, such as Creutzfeldt-Jakob disease [15], and many other degenerative and neurodegenerative disorders. It is also related to various metabolic disorders, including Gaucher’s disease [16] and Type 2 diabetes [17], as well as cardiovascular diseases, such as atherosclerosis [18] and even cancer [19,20]. Currently, there are no known cures or effective treatments for these progressive disorders. Therefore, understanding how proteins fold may enhance disease management and improve health [21].

To better explain the protein folding process, Levinthal introduced Levinthal’s paradox, which determines that finding the native folded state of a protein by a random search can take longer than the age of the universe. This is explained by the estimation of the average geometrical arrangements that a human-sized protein could adopt 10^143^, and the trillions of folds per second among all possible configurations that a random search requires [22]. Therefore, there are likely rules and gates, which we still do not fully understand, that direct the folding process of a protein to its desired native state. Another approach to predicting protein folding applies artificial intelligence (AI). For example, it was recently published that an AI program, AlphaFold2, scored above 90 (out of 100) for around two-thirds of the proteins in a challenge called Critical Assessment for Structure Prediction (CASP), which measures the degree to which a computational program predicts the final protein structure based on a sequence of amino acids [23]. While this is exciting progress, the accuracy of the prediction is still not high enough for one-third of the proteins, and the AI program does not reveal the mechanism or rules that guide protein folding.

In this section, we compare the folding principles, the folding constraints, paper medium and the medium which proteins form, as well as folding energy.

### 2.1. Folding Principles

Different small proteins fold in a two-state process [24], and some folding processes in proteins proceed in parallel, e.g., two folded structures connect to create a complex protein. Two-state proteins are proteins that fold directly to the native state without transition to other intermediate stable states. The transition theory, which is a simple model that describes the folding process, assumes that the two-state protein has a third intermediate unstable transition state of the partly folded structure. Similar processes can be found in origami; for example, in modular origami, pieces are folded and then assembled to create a complex origami model, and as shown in Section 2.1.2, there are origami models that have only two stable states, folded and unfolded. However, since, in some cases, sequential folding of amino acid chains and polypeptides is fundamental for the creation of proteins, and since sequential folding of paper drives the creation of origami models, it is the focus of the comparison that is presented herein.

#### 2.1.1. Folding Process of Proteins

One of the major parameters that govern protein folding is the folding code that is partially embedded in the amino acids [25]. Each amino acid is composed of four main chemical elements: carbon (C), hydrogen (H), oxygen (O), and nitrogen (N), and are connected via peptide bonds, which are covalent bonds formed between the carboxylic acid group of one amino acid and the amine group of another amino acid. In addition, side chain (R group) chemistry is critical to protein structure (and function, not discussed herein), as these side chains can bond with one another to hold a protein in a certain shape or conformation. Typically, interactions between different side chain groups in native states involve Van der Waals contacts. In general, there are 20 different amino acids that differ from each other in their R group. Protein folding can be categorized into four main stages: (i) primary structures that are comprised of the linear amino acid sequence, (ii) secondary structures, which are characterized by repeating patterns of hydrogen bonds that spatially stabilize the structure, for example, alpha-helix (*α*-helix) or beta-sheet (*β*-sheet), (iii) tertiary structures, which are protein subunits that combine secondary structures that are stabilized in spatial relationships, primarily via salt bridges, hydrogen bonds, and disulfide bonds, and by the formation of a hydrophobic core, and (iv) quaternary structures, which are several folded subunits arrangements forming a complex protein. These proteins are usually made up of two or more polypeptide chains (Figure 2), where cross-interactions and hydrophobic/hydrophilic residues (R group) are connected to the main polypeptide chain and can influence the folding course and the final structure of the protein.

#### 2.1.2. Folding Process of Origami Models

While protein folding is driven by the physical arrangement of the amino acid chain and chemical forces, in origami, the blueprint folding code is embedded in a crease pattern (Figure 3C), where folding along crease lines is governed by physical properties, and the direction of the fold determines whether it is a mountain or a valley fold (Figure 3A,B). The crease pattern, together with the folding sequence, is the information that is required to fold an origami model (Figure 3D), and the more folds there are, the more complex and detailed the model is. In the past, origami models were designed via trial and error. However, in recent years, researchers successfully translated the folding principles into a mathematical language and developed various computerized tools that help design crease patterns that can be folded into the desired model [26]. 

Many origami models are folded in sequence and have several stable states (Figure 3C–E). However, some origami models fold in a single step and only have two stable states, folded and unfolded (Figure 3F,G), similar to proteins that fold in two-state kinetics [28]. 

### 2.2. Folding Constraints

In proteins and origami, folding is governed by various parameters, including geometrical constraints. The folding of a short amino acid sequence (peptide) is constrained by the allowed dihedral angles of the amino acids along the chain [29]. Origami folding also depends on the angles between the creases and the dihedral angles between folded paper panels. When certain mathematical conditions are fulfilled, the paper panels between the creases do not bend during folding. The ability to fold an origami model while keeping the panels straight is called “rigid origami”, which can be exploited to design origami-based foldable products from stiff non-paper materials [30].

#### 2.2.1. Protein Folding Constraints

The dihedral angles between the atoms along the backbone of amino acid chains are one of the parameters through which the folding of a protein is analyzed. The orientation of adjacent planes is determined by the restricted values of the torsion phi-angle (*φ*) and psi-angle (*ψ*) about the Cα-C bond and the N-Cα bond, respectively (Figure 4). The omega-angle (*ω*) around the peptide bond between C and N has a double-bond character and is therefore usually 180°. The Ramachandran plot [31] shows the correlation between the dihedral angles (*φ*, *ψ*) of amino acid residues, and the regions of interest for the two main conformations of the secondary structures α-helix and β-sheet are (−60°, −50°) and (−120°, 120°), respectively. For example, in a parallel β-sheet structure, the H-bond angles are (−119°, 113°).

#### 2.2.2. Origami Folding Constraints

Flat-folding origami, which is folding a model to the paper thickness (or the thickness of the folded medium), is the basis for origami and is achieved when several geometric conditions are fulfilled. In a model with several creases, that intersect in a single vertex is folded (Figure 5A), some creases fold to the right and some to the left. Taking the sectors of the panels that fold to one side to be positive and the sectors of the panels that fold to the other side to be negative, the Kawasaki-Justin theorem states that in a flat fold crease pattern, the iterating sum of sector angles (of panels that are folded to the right and sector angles of panels that are folded to the left), should be equal to zero [32]. For example, in a flat-fold with four creases (Figure 5A), the sum of the angles between the creases is:(1)A+B+C+D=0
where *A*, *B*, *C*, and *D* are the angles between the creases. It is noted that since the angle around a vertex is 2*π*, the sum of each side of the equation is equal to *π*:(2)A+C=B+D=π

Folding a model from a rigid non-paper material must fulfill additional mathematical conditions, which enable folding a crease pattern without bending the panels between the creases (Figure 5B). This means that it is possible to design an origami-based foldable model from stiff materials, such as wood and metals. The mathematical conditions for rigid folding encompass the angles between the creases and the dihedral angles between the panels as follows [33]:
(3)$$1-\cos n=\frac{\sin A\sin B}{\sin C\sin D}\left(1-\cos m\right)$$
and
(4)$$1-\cos q=\frac{\sin B\sin C}{\sin A\sin D}\left(1-\cos p\right)$$
where *m*, *n*, *p*, and *q* are the dihedral angles between sheet panels (Figure 5A). Hence, based on Section 2.2.1 and Section 2.2.2 dihedral angles govern the correct folding of both proteins and rigid origami. 

### 2.3. Properties of the Folded Mediums 

It is claimed that the analogy between protein folding and origami is not accurate because proteins are folded from a chain of amino-acid while origami models are folded from paper [34]. Although, at first glance, it seems that these mediums have little in common, similarity emerges when secondary structures and higher proteins are compared with paper. 

The structural arrangement and chemical bonds between the amino acid chains determine the properties of the folded motif. For example, the basic unit of *β*-sheet is the beta-strand (*β*-strand), which comprises a polypeptide chain, typically 3–10 nanometers long. The *β*-strands in the *β*-sheet are arranged side by side, connected by hydrogen bonds. Similarly, paper, which is the acceptable material for folding origami models, is produced from chains of monomeric glucose molecules. The network of lignocellulosic fibers in paper is connected by hydrogen bonding, and the properties of the paper are determined by parameters such as fiber orientation and fiber-to-fiber bonds [35]. Moreover, the overall geometry of parallel *β*-sheets is not planar but rather is pleated, with alternating Cα carbons above and below the plane of the sheet. The bonds between the Cα’s in adjacent strands define planes that are often drawn as a paper map with a folded zigzag shape (Figure 6). Hence, the name “*β*-sheet” is not random as it describes the “paper” shape of the structure with folded planes. 

There are two major classes of *β*-sheets, the parallel *β*-sheet and the anti-parallel *β*-sheet. Parallel *β*-sheets characteristically feature hydrophobic side chains distributed along both sides of the sheet, while anti-parallel *β*-sheets are usually arranged with all the hydrophobic residues on one side. Hence, an anti-parallel *β*-sheet has two surfaces with opposite characteristics. Closing a *β*-sheet by connecting the last *β*-strand with the first forms a *β*-barrel that can act as a structure that biologically constrains objects from passing through, reminiscent of filter paper. The inner side of the *β*-barrel is primarily hydrophilic and lipophobic, while the exterior is hydrophobic and lipophilic. Therefore, the *β*-barrel can anchor itself into a cell phospholipid bilayer and function as a transmembrane pore that enables the free flow of an aqueous solution [36].

### 2.4. Folding Energy

A folding-free-energy landscape is often used to describe the mechanism that governs protein folding [37]. In recent years, researchers have also been studying the folding energy of origami models [38]. A short review of the energy landscapes of proteins and origami models is presented here, with a focus on the sequential folding of proteins. A full comparison of the folding energy of proteins and origami is beyond the scope of this paper, as it requires a comprehensive analysis, and comparison with different models, such as the free energy barrier in two-state proteins [28].

Protein folding is often described as a process that occurs along a funneled glassy landscape, which is defined by free-energy and entropy of the folded conformations. It is possible to draw a 3D free-energy landscape map of protein folding, where the *Z*-axis represents the free energy and the *XY* plane the entropy (Figure 7A). During protein folding, as the number of native interactions between side chains of the folded structure increases, the folding free-energy decreases, and since the structure becomes more stable, the entropy also decreases. Folding stops when a native structure, with minimal free energy and the maximal number of native contacts, is achieved.

Folding-free-energy can explain how some folding motions of amino chains are restricted by covalent bonds and adjournment forces, such as forces that are created by native contacts during the folding process [39]. This forms an energetic barrier, as the amino chain needs to fold against the pulling forces of the formed contacts. Hence, energetic barriers limit the folded structure to specific volumes, favoring only a limited number of consecutive folds [37]. Local free-energy minima that the protein may reach are considered traps since the protein needs to unfold into a previous state before it folds correctly and continues toward the native structure. Hills are transition states between local minima and represent local maxima in the energy surface.

The folding energy in origami depends on the force that is required to transform the paper into a folded model. Refolding along existing creases that obey flat-folding constraints (such as the crease pattern shown in Figure 5A), requires minimal energy. However, combining several origami folds in a loop, i.e., a loop of four origami folds (the crease patterns in Figure 7B), imposes new constraints and therefore requires higher folding energy due to deviation from the prefolded creases and the resistance of the paper to bending and stretching [41,42]. It is possible to create a mechanical model that mimics the paper response in simulations by translating the stiffness parameters of the paper to linear and rotational springs [43]. 

Presenting the folding energy of origami models in a logarithmic form yields a glassy energy landscape with a minimal energy ground state for the desired fold and an exponential number of other states of higher folding energy. However, unlike in proteins, in origami, new bonds are not formed during folding, and energy is only required for moving and turning the paper.

## 3. Protein Structures and Origami Models

Much research has been devoted to understanding how proteins function and interact with their environment, and although proteins are nanometer biomolecular structures, while origami models are generally millimeters to centimeters in size, it has been applied to protein research. For example, origami has been applied to solve the protein folding code [44], and in a limited number of cases, it has been applied to reveal how proteins interact with the environment and how protein structure contributes to the function it fulfills. The challenge in the latter is to identify in origami governing rules, patterns, and models that translate to mathematical and engineering principles relevant to biomolecular structures. 

In previous research, we studied the analogy between protein structures and origami models, through an approach that compared the shape and structural properties [45,46]. The comparison approach involves several steps such as comparison of the unfolded protein structure with the crease pattern of the origami model, analysis of the geometrical patterns, topology analysis of protein domains and origami flaps, and application of mathematical tools such as finite element analysis (FEA), to study and characterize the structure. At this time, only proteins with a structured shape have been compared with folded origami models, and the similarity between the protein structures and origami models, which were studied, is apparent (i.e., Figure 8). In the future, it may be possible to compare and analyze native unstructured proteins with less defined folded paper models, such as crumpled paper.

For example, the CorA magnesium transport complex, which is assembled from three domains, the cap transmembrane domain, the α-helical barrel domain, and the N-terminal domain, was shown to be analogues to an origami model, which is assembled from three known origami models, two Kresling models and a Flasher model (Figure 8) [45]. From the analogy between the protein structure and the origami models, it was possible to learn about the mechanical properties of the protein structure, such as auxeticity vide infra. Based on the analysis, a hypothesis was raised regarding the role of the CorA protein structure in defining its function. We further elaborate on the unique mechanical properties of origami models and the analogues mechanical properties of protein structures in Section 4.

## 4. Mechanical Properties

The ability of a structure to respond to an external force in a desired way depends on its mechanical properties. A structure can be designed to dampen or amplify forces, and a mechanism can be designed to move in a desired way when an external stimulus is applied. Given that proteins are the basic building blocks of life, it is assumed, that through natural selection, protein structures evolved that better serve their function. The following section discusses some of the unique mechanical properties of proteins and origami that result in structures with a high stiffness-to-mass ratio and the ability to withstand high forces.

### 4.1. Tensegrity 

The term tensegrity comes from the combination of the words tension and integrity and describes structures that are in equilibrium between tension and pressure elements: beams that can withstand pressure and tension forces, struts that can withstand pressure forces, and cables that can withstand tension forces. These elements connect at vertexes, where every vertex is in force equilibrium (Figure 9A). The unique construction of tensegrity structures achieves a high stiffness-to-mass ratio and distributes a force that is applied in a specific location to the entire structure, therefore limiting local damage. A balloon, which stretches against filled air, is an example of a tensegrity structure. Another example is spider webs, which are constructed from stretched cables. In this section, we present evidence that supports the following claims: (i) proteins can be modeled as tensegrity structures, and (ii) it is possible to transform some origami models (with non-crossing elements) to tensegrity structures and vice versa. 

#### 4.1.1. Tensegrity in Proteins 

Several studies support the claim that tensegrity is applied at a molecular level, for example in cells [50], and, more specifically, in proteins [51]. One model suggests that the protein structure, which is comprised of amino acid chain of polypeptide, is balanced by a hydrogen bond network [52]. Hence, the amino acid chains with the covalent bonds are the “beams”, and the hydrogen bonds that pull/push one amino acid chain relative to the other are the “cables” and “struts”. For example, in *β*-sheets, the hydrogen bonds between the strands are compressed (Figure 9B), and in α-helix, the hydrogen-bonded residue pairs are under tension. Therefore, the interactions between close residues determine the prestress of the protein structure, which in turn determines the preferred conformation in each folding step [47]. Furthermore, the tensegrity structure in proteins allows them to withstand external and internal forces, and to change the form of the flexible surface.

#### 4.1.2. Tensegrity in Origami 

Tensegrity structures and origami models are correlated and it is sometimes possible to transform from a tensegrity structure to an origami model. For example, it has been shown that spider webs, which are tensegrity structures constructed from tensioned fibers (e.g., cables) [49] (Figure 9E), can be simulated via equivalent origami models. The equivalent model is an origami tessellation model, which is a tiled pattern, that can be folded from a single sheet of paper (Figure 9C). In addition, it is also possible to translate tensegrity structures, with non-crossing elements, into origami models by integrating panels into the structure [53]. The opposite is also true, namely, it is also possible to translate some origami models (e.g., polyhedrons) into tensegrity structures by converting valley folds to cables, mountain folds to struts, and panels to beams (Figure 9D) [48].

Importantly, following the above, it may be possible to use an intermediate tensegrity model, to transform from origami models to de novo protein structures, and to translate structural principles of proteins to origami models to study their mechanical properties. The latter is important since it is a lot harder to apply forces and track the reaction of nanoscale proteins than it is to study origami models that are generally in a centimeter-scale.

### 4.2. Auxeticity

Poisson’s ratio, ν, is the ratio of the deformation of a structure under load, along its axes. In most materials in nature, and in keeping with mass conservation principles, an increase in length along one axis is accompanied by a decrease in length along the other axes. However, there are structures that when stretched increase their length along axes that are perpendicular to the applied force. These structures have a negative Poisson’s ratio (NPR), which is also termed “auxetic”, meaning in Greek “that which tends to increase”. In this section, we present how auxeticity appears in protein structures and origami models.

#### 4.2.1. Auxeticity in Proteins

Auxetic structures are constructed with gaps that allow the structural elements to change their relative position by closing/opening the gaps. Changes in the position of elements result in contraction/expansion of the structure. Proteins that are also constructed with gaps between atoms, amino-acid chains, and secondary and tertiary structures, can conform by changing their volume. As mentioned above, we have shown, through analysis of protein conformations, that the CorA protein is analogous to an assembly of three origami models. An analysis of the mechanical properties of the origami models demonstrated that the CorA mechanism, which functions like a mechanical shutter of a camera, is auxetic. Namely, like the Flasher model, CorA simultaneously expands/contracts in a synchronized motion along two axes, which results in volume modification to fulfill its biological function, i.e., to transport magnesium across the cell membrane [45].

A similar mechanism is found in the mechanosensitive ion channel (MscS), a membrane valve that is designed to relieve osmotic imbalance, allowing a rapid and transient increase in compensatory solute flux out of the cell. When bacteria are exposed to freshwater, the pressure on their membrane is balanced by valves such as the MscS. The MscS is assembled from three domains, and forces on the protein lead to the twist motion of its transmembrane domain. The twist motion opens/closes the opening by simultaneously expanding/contracting along two axes (i.e., *X* and *Y*-axes, Figure 10A,B) [54], and this is similar to the twist motion of the auxetic Flasher origami model (Figure 10C,D).

#### 4.2.2. Auxeticity in Origami

Many origami models, such as the Flasher model (Figure 10C,D), are auxetic. Furthermore, some origami tessellations mentioned in Section 4.1.2 are auxetic and are used for the construction of mechanical metamaterials with desired properties [55,56,57]. The auxetic characteristic is demonstrated through a force analysis, at a vertex, in a tile that constructs an origami tessellation (Figure 11) [58]. Each vertex in the tile is a rotational joint (point), and it is connected to several elements that construct a unit cell, that stretches or shortens, depending on the direction of the external force (for example, vertex *K* in Figure 11B). A joint is in force equilibrium when fully folded or deployed. In both of these states, the square sides are tilted by an angle of +*α* or –*α*, in relation to the angle of the cable, such that the forces at the vertexes along the *Y*-axis are in equilibrium. Similarly, the forces along the *X*-axis at the vertexes are in equilibrium, i.e., the tile changes between two states, from state B (Figure 11C) to state A (Figure 11B), and since the dimensions of the tile increase along both axes, we can deduce that it is auxetic [56]. The length of the tile along the *X*-axis increases by:(5)ΔW=2h(sinα)
where *h* is the height of the square, α is the rotation angle of the square about the *Z*-axis and ΔW is the difference in length along the *X*-axis. The length along the *Y*-axis increases by:(6)ΔH=2w(sinα)
w is the width of the square, and ΔH is the length difference along the *Y*-axis.

## 5. Summary and Conclusions

In this study, we presented evidence for the resemblance between proteins and origami. The examples and analyses, which are described here for the first time, provide a unique view of the analogy from various perspectives. The main conclusions are summarized in Table 1.

As detailed in the table, there are similar aspects in the folding of proteins and origami, such as the folding principles (e.g., both require a specific process that is governed by a similar mechanism), folding constraints (e.g., limitation by dihedral angles), and the folded medium (e.g., both are made by polymerization of long chains). The similarity is also noted in the shape and unique mechanical properties of folded protein structures and origami models, and it is possible to draw parallel lines between cases of misfolding.

Results from studies on proteins and origami also provide strong evidence in support of the equivalence between proteins and origami models. Therefore, origami is an attractive model that can provide a unified way to explain different aspects of proteins forming, the folded structure, and its mechanical properties. For example, since proteins fold quickly and it is hard to achieve detailed visual images and track their exact folding process, a simulated or physical origami model, which is similar to a protein structure (or complex), can be applied to explore the protein folding process. The origami model can be folded and unfolded many times to follow various folding sequences that result in different models, and studying the folding process and the folded and misfolded models can help analyze protein folding.

Understanding origami and protein analogy can provide new insights into protein folding and function, and since proteins’ misfolding has emerged as a process of enormous medical relevance, the origami technique may have an impact on disease and medicine research. Furthermore, the ability to translate origami models to protein structures can further guide the design of de novo proteins and nanomaterials with desired properties.

## Figures and Tables

**Figure 1 biomolecules-12-00622-f001:**
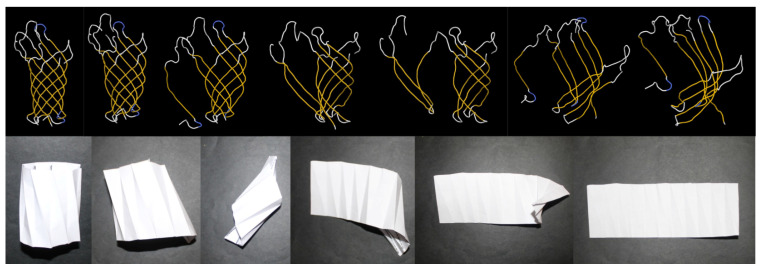
Folding processes of proteins (**top**) and origami (**bottom**).

**Figure 2 biomolecules-12-00622-f002:**
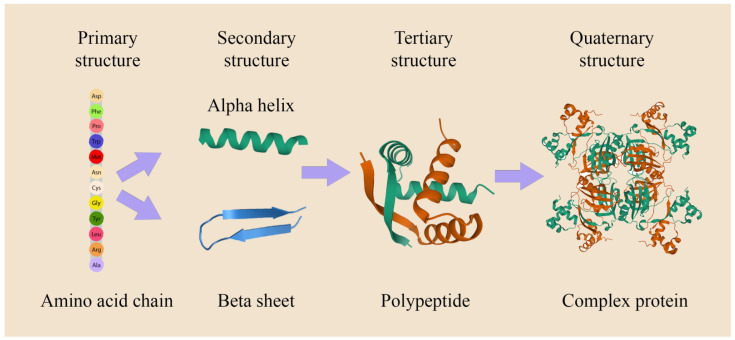
Hierarchy of protein structures.

**Figure 3 biomolecules-12-00622-f003:**
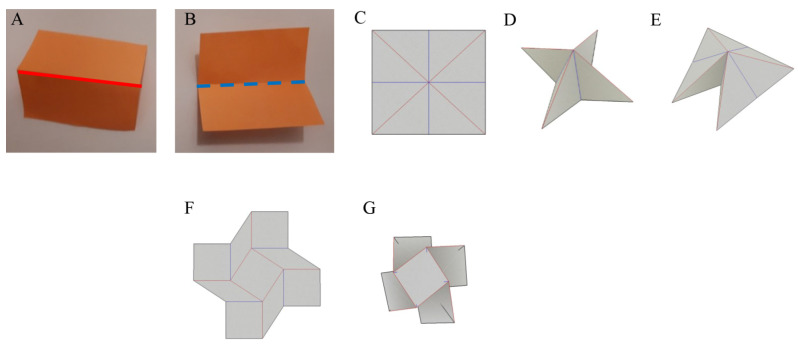
Folding variations and concepts in origami models. Folding variation: (**A**) mountain fold, and (**B**) valley fold. The “double square” origami model with several stable states: (**C**) unfolded model, where the mountains in the crease pattern are marked in blue and the valleys in red, (**D**) a model that is folded from the crease pattern (**E**) a second folded stable state of the origami model. A “square twist” origami model, which is often incorporated in origami tessellation models, with only two stable states: (**F**) unfolded model, and (**G**) folded model (simulated in Freeform Origami software [27]). The model folds synchronically from unfolded to folded state.

**Figure 4 biomolecules-12-00622-f004:**
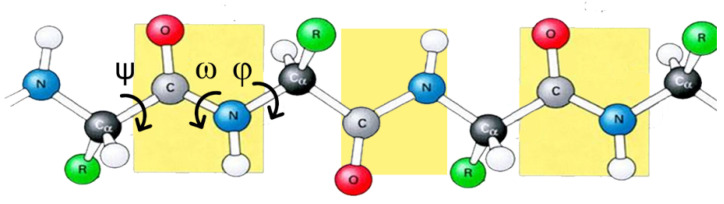
Dihedral angles between the atoms along the backbone of peptide chain.

**Figure 5 biomolecules-12-00622-f005:**
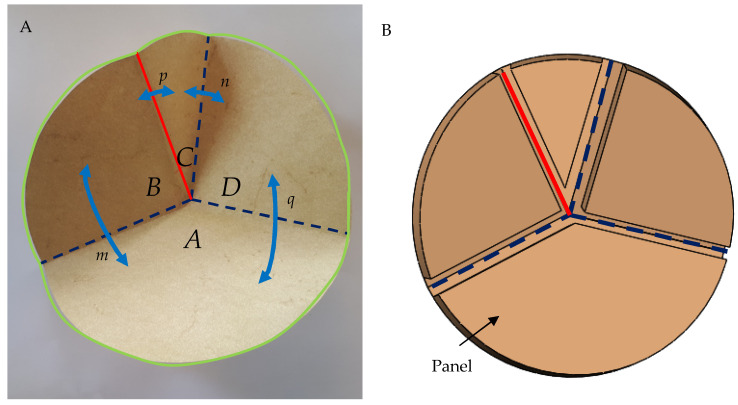
Origami model: (**A**) A half-folded paper model showing the scheme of the angles between the creases (*A*, *B*, *C*, and *D*) and the dihedral angles (*m*, *n*, *p,* and *q*) of a fold with creases that intersect in a vertex. The valley and mountain folds are marked by blue dashed lines and continuous red lines, respectively. (**B**) Rigid origami—the panels are thick and do not bend during folding.

**Figure 6 biomolecules-12-00622-f006:**
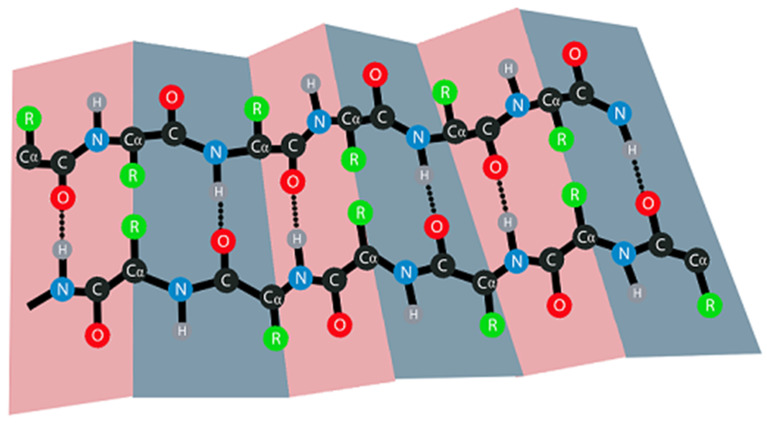
Scheme of the folding in a parallel *β*-sheet. The folded “paper” planes in the model are created by creases (marked in pink and grey), that are defined by the bonds between the C*α* atoms of the two *β*-strands (marked in black structure).

**Figure 7 biomolecules-12-00622-f007:**
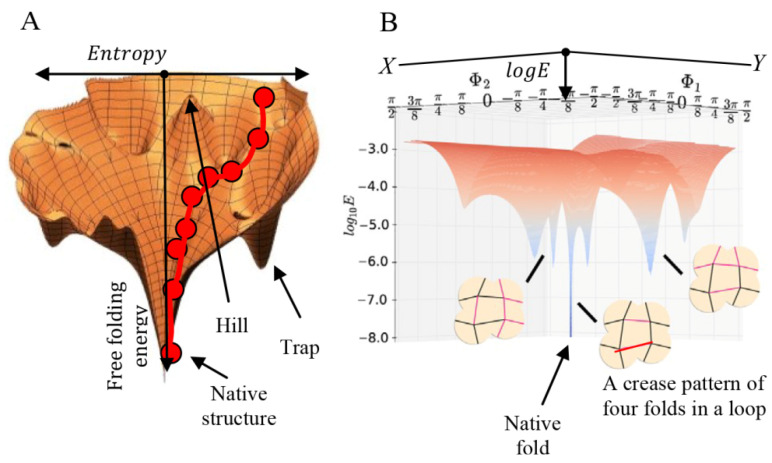
The energy graph of protein and origami folding. (**A**) Energy landscape of protein folding, including hills, traps, and energy barriers. Red dots represent folded states along the folding process. (reprint from [40] with permission from Elsevier). (**B**) Energy landscape of paper refolding. The graph shows the glassy landscape of a loop chain of creases with connected vertexes. $\Phi_{1}$ and $\Phi_{2}$ are the eigenvectors of the angles of the four creases at each of the vertexes of the crease pattern. The *Z*-axis is the free energy, which is required to refold a model when deviating from the prefolded creases. The red (continuous) and blue (dashed) lines in the crease patterns represent mountain and valley folds, accordingly (reprint from [41] with permission from APS physics).

**Figure 8 biomolecules-12-00622-f008:**
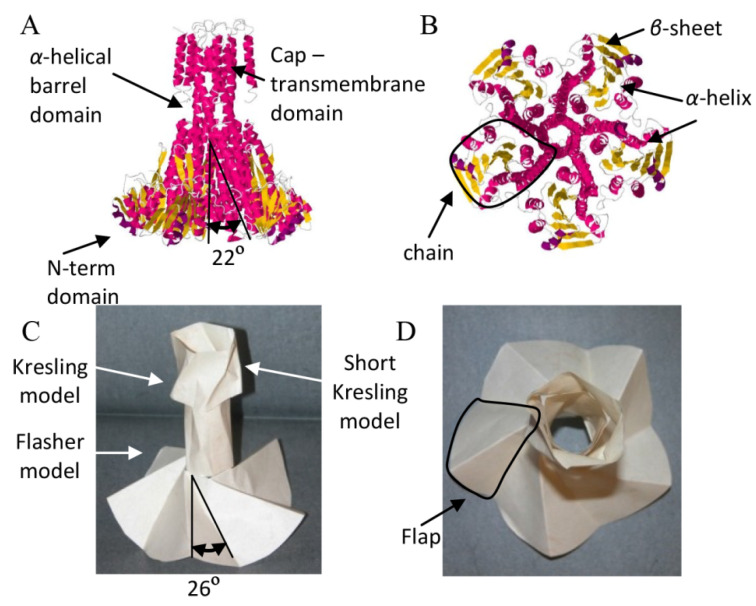
CorA protein structure and analog origami models (reprint from [45] with permission from Elsevier). Protein structure: (**A**) side view of the space-filling model of the CorA protein structure, with cartoon marks for the *β*-strands (yellow) and *α*-helices (red) (PDB: 2IUB), and (**B**) top view of the CorA protein structure, with the black line marking one of the chains. Origami model: (**C**) side view of the origami model that is analog to the CorA protein structure, composed from one Flasher model and two Kresling origami models, and (**D**) top view of the analog origami model, with the black line marking one of the flaps.

**Figure 9 biomolecules-12-00622-f009:**
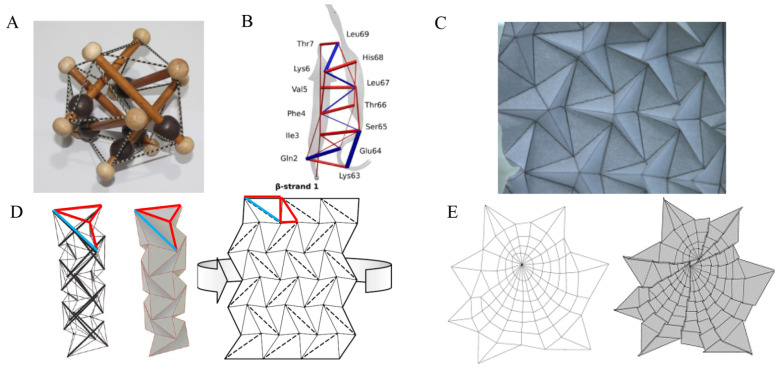
Tensegrity structures. (**A**) A tensegrity toy. The beams are connected to cables, which keep the structure under tension. (**B**) Protein structure: mean inter-residue forces for *β*-strand 1 in a *β*-sheet protein structure. The blue lines indicate pressure, and the red lines tension (Reprint from [47] with permission from PLOS Computational Biology). Origami models: (**C**) A folded model of an origami tessellation. (**D**) 3D tensegrity model (Left), its folded origami polyhedron analog (Center), and the polyhedron crease pattern (Right) (Reprint from [48] with permission from Heldermann Verlag). (**E**) Spider web tensegrity (Left) and its folded origami model analog (Right) (Reprint from [49] with permission from CRC Press).

**Figure 10 biomolecules-12-00622-f010:**
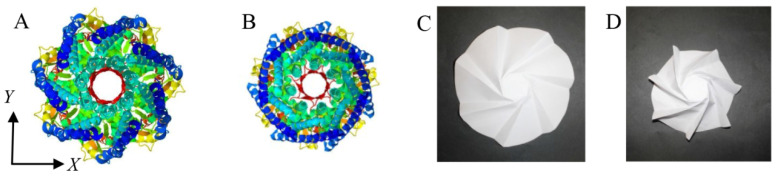
Auxetic iris-like mechanism of pore dilation of a mechanosensitive ion channel (MscS) and Flasher origami model. Protein structure: (**A**) open conformation of the MscS (PDB: 4HW9), and (**B**) closed conformation of the MscS (PDB: 2VV5). Origami model: (**C**) open state of the Flasher origami model, and (**D**) closed state of the Flasher origami model.

**Figure 11 biomolecules-12-00622-f011:**
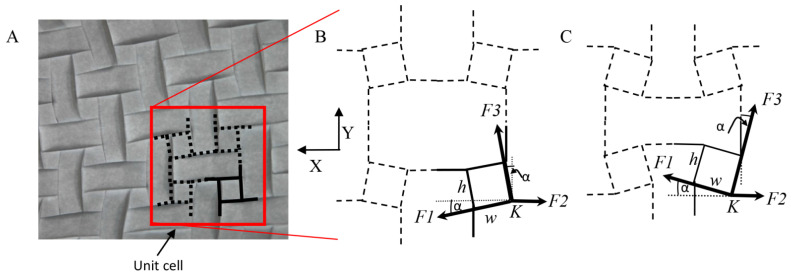
An auxetic origami structure: (**A**) A folded origami tessellation model. Rotation of the unit cell (marked by black lines) causes the pattern to simultaneously contract/extract. (**B**) Unfolded and (**C**) folded states of a unit cell of a tessellation structure (*F1*, *F2*, and *F3* are the forces at the vertex *K*).

**Table 1 biomolecules-12-00622-t001:** Comparison of proteins and origami models.

Topic	Sub-Theme	Similarities
Folding principles of proteins and origami models	Principles (Section 2)	Protein and origami folding are processes that involve sequential folding and assembly of folded subunits, through a series of steps to create a structure. **Protein folding:** Various structures can be folded from a single chain of amino acids and then assembled with other folded subunits. **Origami folding:** Many origami models can be formed from a single sheet of paper. In modular origami, folded models are assembled into complex origami models.
		Protein and origami misfolding cause undesired results. **Protein misfolding:** Misfolded proteins are responsible for various diseases (e.g., Alzheimer’s disease). **Origami misfolding:** Undesired origami models are trashed.
	Folding processes (Section 2.1)	Protein and origami folding require a specific process. **Protein folding:** The folding process of proteins is driven by the sequence of atoms in the amino acid chain and chemical forces. **Origami folding:** The process is based on a crease pattern and driven by physical forces.
	Folding constraints (Section 2.2)	Dihedral angles between atoms and between folded origami panels restrict the folding. **Protein folding:** Dihedral angles between atoms control the amino acid chain folding. **Origami folding:** Rigid-folding origami is obtained by imposing specific mathematical relations between crease angles and dihedral angles of the folded panels.
	Folded mediums (Section 2.3)	Proteins and paper origami are both constructed from polymerized fibers. **Protein folding:** Proteins are constructed with secondary structures, such as β-sheets, which are comprised of several β-strands that are connected by hydrogen bonds. In other proteins, interactions between different side chain groups, typically involve Van der Waals contacts. **Origami folding:** Paper is constructed from polymerized fibers whose orientation and number determine the paper properties. Comment: Although origami models are mainly made of paper, it is also possible to create paperless origami models.
Structure and shape of proteins and origami models (Section 3)		**Proteins & Origami**: Some protein structures and origami models are analogues. For example, β-barrel and Kresling origami model.
Mechanical properties	Tensegrity (Section 4.1)	Tensegrity systems appear in proteins and in origami. **Protein structures:** Cells and proteins are molecular tensegrity systems (e.g., the amino acid chains are the beams, and the hydrogen bonds are the struts and cables). **Origami models:** (1) Certain tensegrity models (e.g., non-crossing), such as spider webs can be translated to tessellation origami models. (2) Particular origami models can be translated to tensegrity structures (e.g., polyhedrons).
	Auxeticity (Section 4.2)	There are proteins and origami models that open/close synchronically. **Protein structures:** Some proteins are auxetic (e.g., the N-terminal domain in the CorA protein). **Origami models:** Many origami models are auxetic (e.g., tessellation-based origami models and the Flasher model).

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
