# Peer review of "How Similar Are Proteins and Origami?"

_biomolecules, 2022, doi:10.3390/biom12050622_

Round 1
Reviewer 1 Report
The authors discuss and review the issue of the possible similarity between protein and origami structure. As far-fetched as the premise may look, the authors do a very good job in highlighting a series of similarities between the two object kinds. I find the manuscript nicely written, and its reading refreshing even for an hardcore protein scientist with little or no knowledge about origami.
I only have a couple of comments the authors need to address
- lines 118 and only entry in column 3 in Table 1: the authors seem to imply that bonds are formed between different side chain groups in protein native states upon folding; this is not true for covalent bonds, with the only exception of disulfide bridges among cysteine pairs, whereas hydrogen bonds between side chain groups are not that common; the typical interaction between different side chain groups in native states involve Van der Waals non covalent contacts. Please correct.
- line 496; it is not clear what the authors mean by "follow various sequences". I guess "various sequences of steps along the folding process" or something along those lines. Please explain better.
and a couple of other minor comments
- since the authors refer to spider web as an example of a tensegrity structure; it may be helpful to observe that spider silk itself is made up of fibrous proteins
- some fraction of protein structures are characterized by topological complexity, such as knots (Taylor doi:10.1038/35022623), links ( https://doi.org/10.1073/pnas.1615862114) and entangled loops (Baiesi et al https://doi.org/10.1038/s41598-019-44928-3); I am curious whether a similar properties can be found in origami structure as well
Author Response
We would like to thank the Reviewer for taking the time to provide us with a thorough review and positive feedback. We did our utmost to address the below concerns of the reviewers as adequately, and in as detailed a manner, as possible. We believe this manuscript is improved, and submit it for publication in Biomolecules. In the attached file we provided is a point by point response (in blue) to the reviewers’ comments (brought verbatim in black).

Reviewer 2 Report
The authors introduce origami to model the properties and folding of proteins. They describe both the similarities and differences between paper origami and proteins.
Proteins are complex molecules and understanding their structure, dynamics, and associated functions is a challenging task. As a result, numerous simplified models have appeared that more or less successfully mimic natural proteins while providing the opportunity to obtain some information thought to be valuable in understanding the physics behind folding or protein function.
It is from this perspective that I understand the research presented, which attempts to compare the properties of proteins to the fully mechanical model represented by origami. Although it is a very crude model, they were able to find some similarities between these groups.
More than the similarities, I am concerned about the differences in using origami as a model. They compare the completely homogeneous structure of paper (origami is macroscopic) to proteins, which are not homogeneous on any scale. The fact that different AA have different side chains is the origin of protein folding and that different protein structures have different functions. Protein folding is driven by physical forces emanating from different AA with different side chains. I also do not see the fact that the paper structure at the atomic level is based on the hydrogen bonds as being related to the shape of origami. In any case, mixing the atomic level of proteins or the paper structure with the macro level of the origami property itself does not seem convincing to me, even if some kind of mathematics is introduced.
Also, in introducing free energy into the manuscript (equation 5), I overlooked an important contribution by enthalpy. Besides entropy, this part of free energy is of extreme importance mainly due to the changes in the formation of the H-bonds between the solvent (water) and the AA side chains, especially when the difference in free energy between the folded and unfolded states is very small.
In the introduction (which should be shortened considerably), the authors claim that AI is very effective in predicting protein structure. This is of course a very important result, but to get dipper into the folding mechanism or to understand and even more important to prevent the misfolding, the exact physical picture (forces) should be known. And I am afraid that this kind of modelling would not significantly improve the current knowledge of the underlying protein physics.
Although I think that the proposed model has some weaknesses, I recommend this paper for publication. It could be a unique way to identify some new data in the field of protein research.
Author Response

(The authors gave the same response as above.)
